# Using a Flexible IoT Architecture and Sequential AI Model to Recognize and Predict the Production Activities in the Labor-Intensive Manufacturing Site

**Cadmus Yuan *** , **Chic-Chang Wang, Ming-Lun Chang, Wen-Ting Lin, Po-An Lin, Chang-Chi Lee and Zhe-Luen Tsui**

Department of Mechanical and Computer-aided Engineering, Feng Chia University, Taichung 407802, Taiwan; chicwang@fcu.edu.tw (C.-C.W.); mlchang@fcu.edu.tw (M.-L.C.); wtlin.mcae.fcu@gmail.com (W.-T.L.); D0539570@o365.fcu.edu.tw (P.-A.L.); cclee.mcae.fcu@gmail.com (C.-C.L.); gerald10719@gmail.com (Z.-L.T.)
* Correspondence: cayuan@fcu.edu.tw; Tel.: +886-939-873-055

**Abstract:** Under the pressures of global market uncertainty and rapid production changes, the labor-intensive industries demand instant manufacturing site information and accurate production forecasting. This research applies sensor modules with noise reduction, information abstracting, and wireless transmission functions to form a flexible internet of things (IoT) architecture for acquiring field information. Moreover, AI models are used to reveal human activities and predict the output of a group of workstations. The IoT architecture has been implemented in the actual shoe making site. Although there is a 5% missing data issue due to network transmission, neural network models can successfully convert the IoT data to machine utilization. By analyzing the field data, the actual collaboration among the worker team can be revealed. Furthermore, a sequential AI model is applied to learn to capture the characteristics of the team working. This AI model only requires training by 15 min of IoT data, then it can predict the current and next few days' productions within 10% error. This research confirms that implementing the IoT architecture and applying the AI model enables instant manufacturing monitoring of labor-intensive manufacturing sites and accurate production forecasting.

**Keywords:** human activity recognition; labor-intensive industry; IoT; Cloud database; AI model; neural network model; long-short term memory

## 1. Introduction

Most labor-intensive manufacturing focuses on the mass production of large varieties of types, colors, and sizes, complicated supply chains, and different manufacturing processes. These production sites are often occupied with thousands of workers and dozens of differentiated manufacturing lines for multiple procedures and products. The manufacturing goals often fluctuate due to the rapid change of the market needs, the unstable supply chains, and the unpredictability of the manufacturing site. Under the global trend of manufacturing digitalization, good scheduling and production predictability is critical for business success.

Scott [1] studies the changing global geography of labor-intensive industries and upgrading in industrial agglomerations, and many quite successful cases involving the transition from simple assembly to full-package manufacturing involving new technologies. To maximize the efficiency of labor-intensive manufacturing, the literature applies mathematical tools to optimize production performance. Rafiei and Ghodsi [2] studied the dynamic cell formation problem (DCFP) with human-related issues to optimize production cost and labor utilization. Bagheri and Bashiri [3] proposed a mathematical model for cell formation, operator assignment, and inter-cell layout problems. Egilmeza et al. [4] studied the stochastic skill-based manpower allocation problem, a daily manufacturing

issue. Tang et al. [5] studied the efficiency of the stochastic two-sided assembly line and optimization methods. Kuo and Liu [6] studied the operator assignment problem involving inter-cell manpower transfer and applied the mathematical model to field application. Afshar-Nadjafi [7] summarized the multi-skill workers' scheduling problem for flexible manufacturing.

Field application is never ideal because multiple manufacturing limitations and constraints are encountered, such as short new product introduction time, supply chain instability, and high turnover rates of experienced workers. Although the labor-intensive production lines tend to adjust themselves to fulfill these constraints under acceptable manufacturing efficiency due to the long-time workers' interactions and management experience, accurate production capability is often challenging to obtain. Industrial engineers cannot measure the massive manufacturing information from all production lines simultaneously, therefore, reliable and automatic information collection methods and corresponding analysis methods are required.

Jazdi [8] studied the cyber-physical integrated system and its application for Industry 4.0 and identified the importance of the internet of things (IoT) service. Khaleel et al. [9] described an IoT platform and the development prototype of the functional blocks to realize the business. Kokuryo et al. [10] presented an innovative methodology for value co-creative manufacturing with an IoT-based smart factory. Rezaei et al. [11] developed an IoT-based framework for supply chain performance measurement and real-time decision-making. Kolberg et al. [12] proposed an Industry 4.0 way to realize and visualize lean production concepts by wireless technology. Moreover, Gladysz and Buczacki [13] reviewed the applications of wireless technologies in support of lean manufacturing tools.

The IoT installation should obey the local labor law, avoid influencing the workers' regular jobs, and respect the employee's personal privacy. Harja et al. [14] recorded the information of the working station instead of the workers' activity. In the smart labor-intensive factory paradigm proposed by Kim and Moon [15], the manufacturing information collectors, including the RFID on the products and the digital readers, are applied.

Moreover, these time-dependent data collected from the IoT system should be further analyzed and applied. Babiceanua and Seker [16] reviewed virtualization and cloud-based services for manufacturing systems and the use of big data analytics for manufacturing operations planning, addressing the appropriateness of "what-if" scenarios. Oberdorf et al. [17] applied the data-driven approach to facilitate the digital transformation of a medium-size company and increase the business value. Moreover, Yuan et al. [18,19] reported that time-dependent industrial data could be modeled by sequential machine learning techniques, such as recurrent neural network (RNN) or long short-term memory (LSTM).

This paper proposes a flexible IoT architecture for field data collection and cloud-based data storage for the labor-intensive industry. Moreover, applying the IoT data, two AI models for the machine/labor utilization and production prediction are established. Both the IoT architecture and AI modeling techniques are realized in the application field of a labor-intensive footwear manufacturing site.

This paper is organized as follows: The fundamental scientific issues, application field requirements, and the literature reviews were described in the first Section 1. The following Section 2, provides the IoT architecture design and the basic theoretic approaches of the AI models. The Section 3 describes the implementation of the IoT system in the field. Sections 4 and 5 described the IoT data cleaning principle and the AI modeling procedure and results. The conclusion of this paper is given in the last section.

## 2. Methods

### 2.1. IoT Architecture

Figure 1 shows the IoT architecture applied in this research. Besides the hardware and software integration, the application and security of the data are also considered. The wireless sensors (Figure 1b) are attached to the actual machines (Figure 1a) to measure their physical response, including vibration, temperature, and current. Because these machines

might be moved around to obey the new product introduction, a wireless connection (Figure 1c) is required. Hence, the preliminary sensor data would be processed by the sensors' CPU to filter the measurement noise and to abstract the critical information before the transmission to the wireless hub. Note that, among major wireless technologies for production efficiency improvement reviewed by Gladysz and Buczacki [13], Wi-Fi following the IEEE 802.11 set of standards (2.4–5.2 GHz) was applied. Wi-Fi was available in the application field, and the transmission distance was approximately 5–15 m. Moreover, more than the radio frequency identification (RFID) component that has been often applied for lean improvement, [13] this research applied a vibration sensor to acquire more information to enable the production prediction.

Furthermore, a cloud-based database was applied because the data are from multiple sites of labor-intensive factories in different countries. To improve basic IT security, two different databases (Figure 1d) were designed. One of the databases stores the time-dependent information of each sensor, and the other one stores the correlation table of the sensor and the machine. Local managers can perform the lean production improvement by considering the real-time machine utilization data. The operation center can build the production forecasting model by accessing the cloud database (Figure 1e).

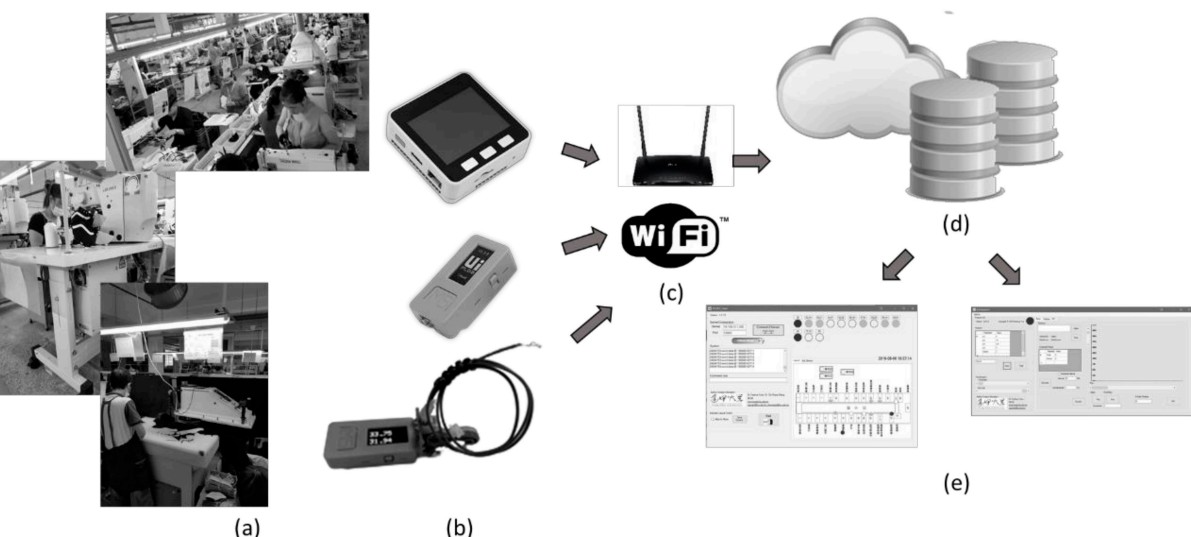

**Figure 1.** The IoT architecture, including (**a**) the field; (**b**) sensors with the preliminary information processing capability; (**c**) secured wireless hub; (**d**) cloud-base database and (**e**) the end application.

### 2.2. AI Modeling Methods

2.2.1. The Single Machine Utilization Model

As mentioned previously, the cloud database stores the time-dependent physical responses of each sensor. With careful selection of the sensor and design of its installation condition and working frequency, a clear difference between the working and non-working response was achieved, as indicated in Figure 2. This research applied two abstracting models to convert the sensor signal to machine utilization, including the threshold and neural network methods.

Referring to Figure 2, the threshold method defines the sensor response at time $t$ as $f_t(t) = \begin{cases} D_k \\ d_l \end{cases}$, where $D_k$ and $d_l$ are the responses during the working and not working period, respectively. We define the threshold function $F$ as

$$F(f_t) = \begin{cases} 1, & f_i, f_{i-1}, \ldots, f_{i-n} > D_{min} \\ 0, & f_i, f_{i-1}, \ldots, f_{i-m} < d_{max} \end{cases}, \tag{1}$$

where $D_{min}$ and $d_{max}$ are the minimum and maximum of the working and non-working responses, respectively. The $n$ and $m$ are the accumulated time that the response value exceeds the thresholds. Equation (1) is often applied to sensor parameter adjustment and the labeling tool for neural network learning.

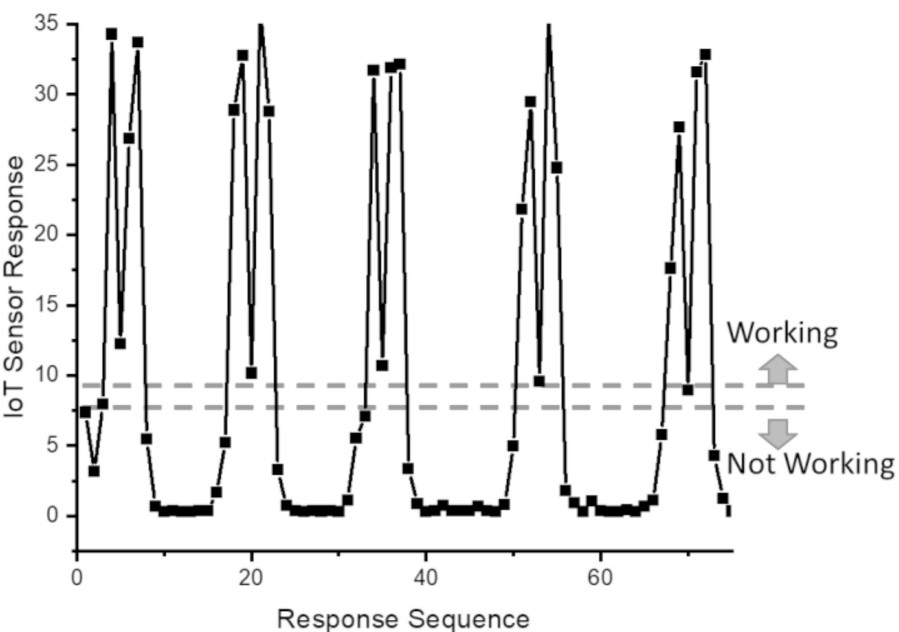

**Figure 2.** A typical response of the sensor that is attached to a semi-automatic machine.

In reality, missing data occurs frequently and it limits the application of the threshold model shown previously. A neural network model with 5 inputs, including the machine status and the sensor output of the current and the previous three-, two-, and one-time steps, is proposed to capture the machine utilization, as illustrated in Figure 3. This neural network architecture is expected to provide a real output, but the working status is discrete (working/non-working). Hence, a softmax function is applied to convert the real number from the neural network to the machine's working status. Moreover, the machine status will become the input for the next timestep. The rectified linear unit (ReLU) is applied as the activation function. The selection of the ReLU among the common activation functions, such as Sigmoid, Tanh, etc., is due to the balance of the learning efficiency and accuracy.

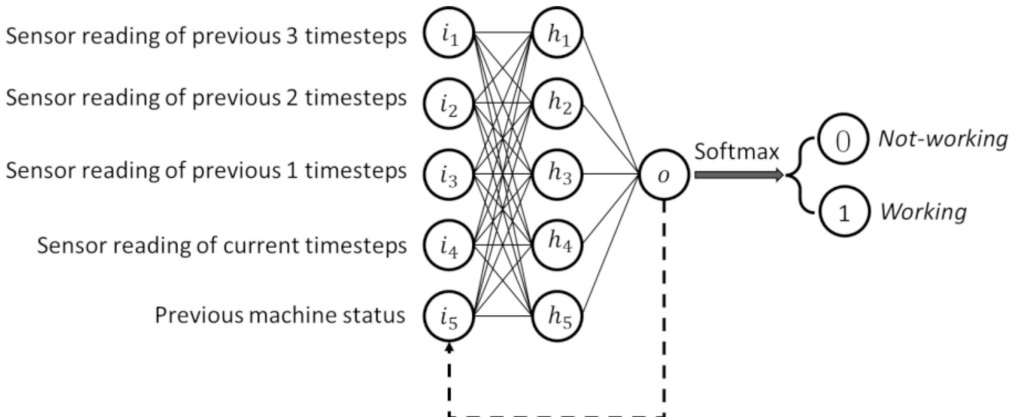

**Figure 3.** The neural network structure to capture the single machine utilization.

2.2.2. The AI-based Sequential Neural Network for Production Line Modeling

From the application field experience of industrial management, there are strong and complex interactions between the up and downstream machines in the production line.

For example, human influence is essential because most of the machines in labor-intensive production are operated by laborers. Hence, machine utilization will be impacted by human activities. Therefore, it is not feasible to directly combine multiple single machine utilization models, as indicated in Figure 3, together and form the AI machine utilization prediction model.

A sequential neural network model is proposed to predict the production line utilization based on machine utilization, as illustrated in Figure 4. Considering the production line model shown in Figure 4a, five key parameters have been abstracted, including "Input(t)", "Stock(0)", "Machine Status(t)", "Stock(t + 1)", and "Output(t)". The sequential manufacturing process is discretized by a fixed timestep that can capture every production line's parameter change.

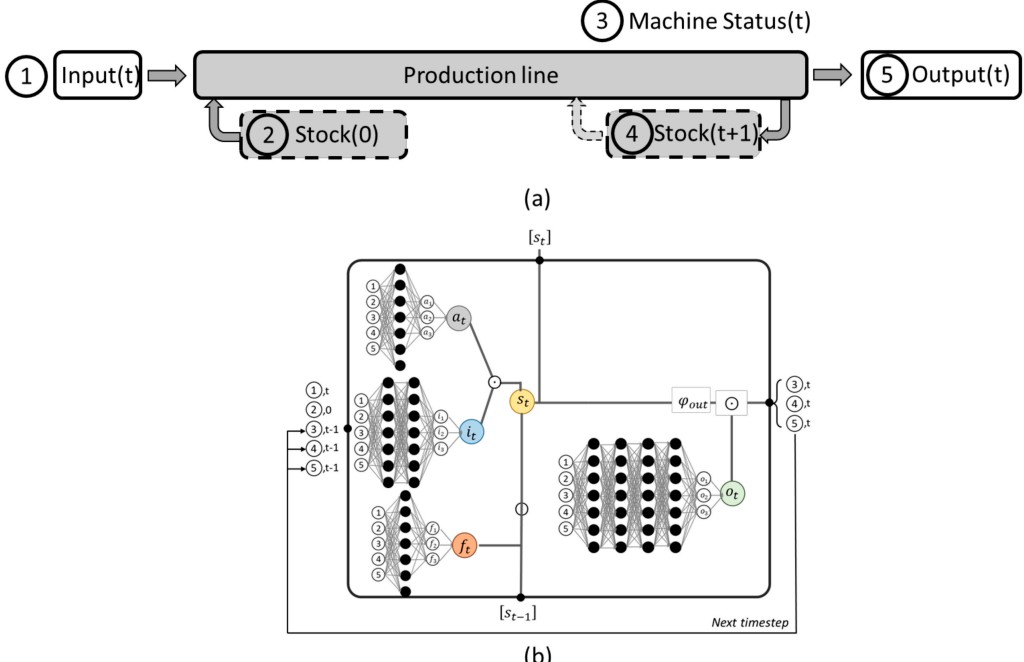

**Figure 4.** The machine learning model for production prediction (**a**) an illustration of the key parameters of the production line; (**b**) the LSTM model.

The "Input" refers to the input material to the production line at time *t*, while "Stock(0)" refers to the unprocessed materials at the beginning. Due to the current "Machine Status", the production line can transform the materials, including the "Input(t)", "Stock(0)", and "Stock(t)", to "Output(t)" or "Stock(t + 1)". To predict the manufacturing status of the next time step $t + 1$, the "Stock(t)", "Machine status", and "Output" of the previous timestep, together with the current "Input(t)" and fixed "Stock(0)", will be used as input vectors to the model.

A sequential neural network model is applied to represent the operation characteristics of the production line. The gate-network long short-term memory (gnLSTM) concept [18] is applied, as illustrated in Figure 4b. The LSTM-based approach was selected because the production line has a short-term memory effect, for instance, the unprocessed stock in the line. Different from the vanilla LSTM, gnLSTM frees the LSTM gates from scalar to vector [18,20], which is the combination of the "Output(t)", "Stock(t + 1)", and "Machine Status." Note that the vanishing gradient issue remains and we applied many trials to find parameters, including the neural network structure and learning speed, which can provide stable training results. The four gnLSTM gates can be expanded to four neural networks with independent neural network parameters and activation functions. The output of the neural network structure is a vector that is obtained by

$$s_t = a_t \odot i_t + f_t \cdot s_{t-1} \, out_t = \varphi_{out}(s_t) \odot o_t, \tag{2}$$

where $\odot$ is the Hadamard (elementwise) multiplication, and $a_t$, $i_t$, $f_t$ and $o_t$ are the gate network output vectors, as illustrated in Figure 4b. $s_t$ and $s_{t-1}$ are the status vectors of time $t$ and $t - 1$, respectively. $\varphi_{out}$ is the activation function of gnLSTM. The learning of the gnLSTM follows the learning procedure in [18].

## 3. Manufacturing Activities Capture

### 3.1. Activities Capturing System

In the application field of the labor-intensive production site, the manufacturing capability highly depends upon human activities. However, it was not easy to deploy direct human activity recognition systems, such as RFID or visual inspection, due to the personal privacy protection policy of the local production company. Hence, instead of direct human activity measurements, the movements of the manual/semi-automatic machines operated by the workers were measured in this research, as mentioned by Harja et al. [14].

A cost-effective, commercially available integrated sensor module with basic CPU and wireless transmission capability was applied in this research as the basic sensing unit. It was equipped with a three-axis acceleration meter and gyroscope, with the measurement range of $\pm 2$, 4, 6, 8 g and $\pm 250$, 500, 1000, 2000 dps, respectively. The sampling rate reached 4 kHz with a 16-bit resolution.

The raw data measured by the sensor module were noise filtered and temporarily stored in the sensor's memory. The average and maximum values of the raw data were computed after a fixed period. The computed data were then labeled with time. Afterward, the labeled information was transmitted wirelessly. The loading of the IoT communication was significantly reduced by the preliminary processing of the sensor module. Moreover, wireless communication technology resolved the rearrangement of the machine layout due to the rapid production changes.

The sensor modules were installed at the actual manufacturing site, as illustrated in Figure 5. First, the installation of the sensor does not influence the daily work nor interfere with the existing machine wiring. Second, the sensors were installed in the locations with the maximum acceleration during machine utilization. Figure 5a–d are four different automatic sewing machines, and (f) is the high post sewing machine at the same production line. Figure 5g,h are the cutting machines.

### 3.2. Data Abstraction

Although the sensor module setup is accomplished under the principles mentioned previously, the sensor responses of the automatic sewing machines differed from each other due to the installation of the sensors, sewing patterns, operators' ways of working, and machine settings. Taking the automatic sewing machines shown in Figure 5a–d as an example, the sewing pattern, the sensor responses, and the threshold parameters of each machine that measure simultaneously are listed in Table 1.

The cycle times of machines (a)–(d) are listed along with the sensor responses. From Table 1, a clear difference can be identified in the amplitude of the sensor reading from sensors (b) and (d), although the sewing patterns are similar. However, the shapes of the sensor responses are similar.

By a careful comparison of the sensor response and the fieldwork, the threshold parameters were obtained and listed in the last row of Table 1. Referring to Equation (1), we simplified the threshold $d$ as $d = D_{min} = d_{max}$. The series of the last row of Table 1 are the threshold (d), and the accumulated time of starting (n) and ending (m) the work, respectively. Hence, the computation of the machine utility can be achieved by the threshold parameters (Table 1) and Equation (1).

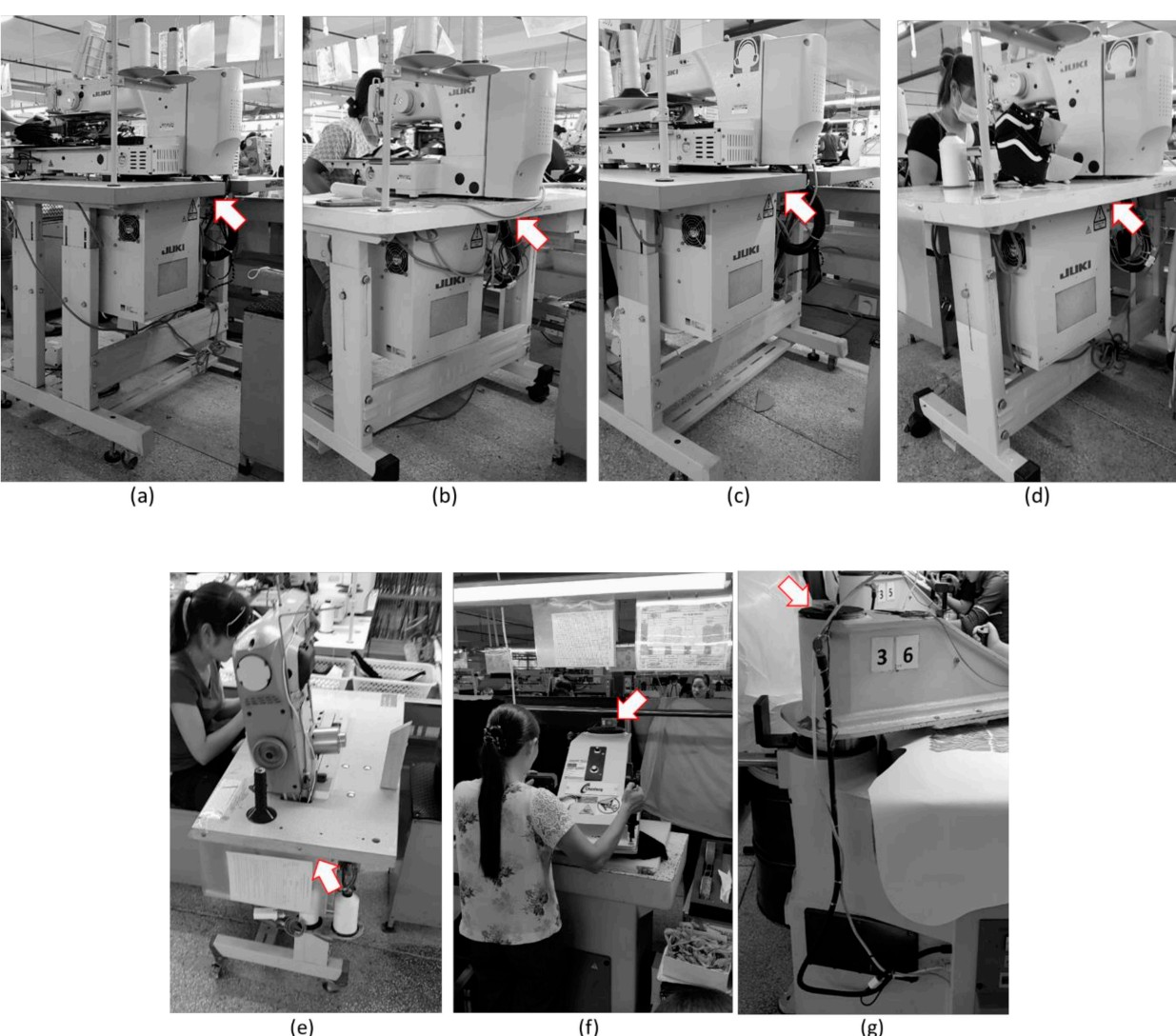

**Figure 5.** The installation of the sensors on the selected machines of sewing and cutting production lines. (**a**–**e**) are the sewing machines with the sensor installation locations (with red arrow), and (**f**,**g**) are the cutting machines.

**Table 1.** The machine utilization conversion from the sensor data by the threshold method.

| Sensor ID | (a) | (b) | (c) | (d) |
|---|---|---|---|---|
| Sewing pattern | | | | |
| Sensor response | | | | |
| Threshold parameters | 2.9 2 1 | 6.5 2 2 | 10 2 2 | 4.5 2 2 |

## 4. IoT Data Pre-processing

### 4.1. The Root Cause of Missing Data

The missing data issue from the IoT architecture shown in Figure 1, has been detected. First, we defined the ideal data count from one particular sensor equals the IoT data resolution (1 Hz) times the total working hours. When the data count from the cloud server was less than 95% of the ideal data count, such a working day was defined as a data-missing day. The root causes of missing data and occurrences of the first IoT implementation month were studied and plotted in Figure 6. The most common cause was the transmission loss from the edge computing (in the factory) to the cloud (outside the factory). The second most common cause was that the data from the IoT could not match the product count from the field. A bad network connection within the factory only occurred once. Hence, Wi-Fi is a reliable technology for data collection for labor-intensive manufacturing sites. Note that the root causes shown in Figure 6 include both technical fault and management negligence because the IoT system was installed in that manufacturing site for the first time.

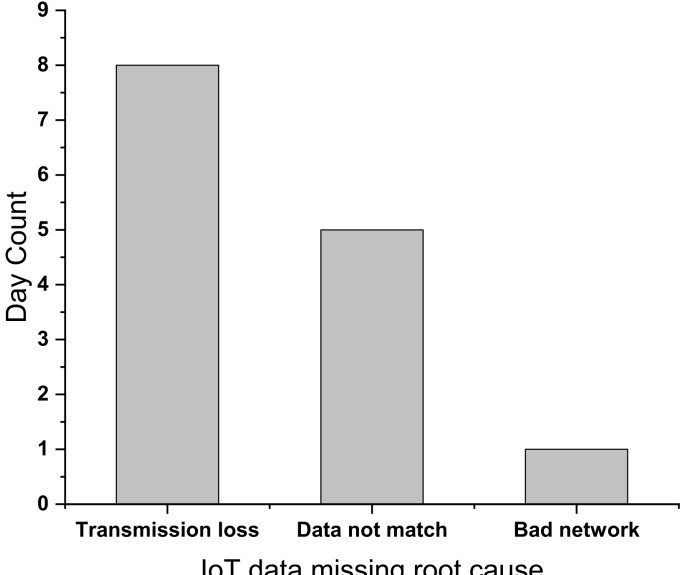

**Figure 6.** The root cause analysis of missing data during the first month of IoT implementation.

### 4.2. Data Pre-processing for the Activity Prediction Model Training

The abovementioned missing data was unavoidable due to hardware limitations, and the threshold method (Equation (1)) became inapplicable. The neural network architecture, shown in Figure 3, was applied to abstracting the characteristics of IoT information. The missing data was filled with the previous signal. These ANN models were trained independently for each machine and updated frequently when the sewing pattern changed and the worker's experience improved. Figure 7 shows the error norm of the ANN machine learning, where 210 data pairs of a continuous operation were selected as the training set under the ReLU activation function and a learning rate of 0.2.

To validate the ANN model, the one-hour data were selected with manual labeling applied. A confusion matrix, as shown by Table 2, was applied to assess the accuracy of the neural network prediction, defined as:

$$accuracy = \frac{TP + TN}{TP + FN + FP + TN}, \tag{3}$$

where the $TP$, $FN$, $FP$, and $TN$ are defined by the confusion matrix in Table 2.

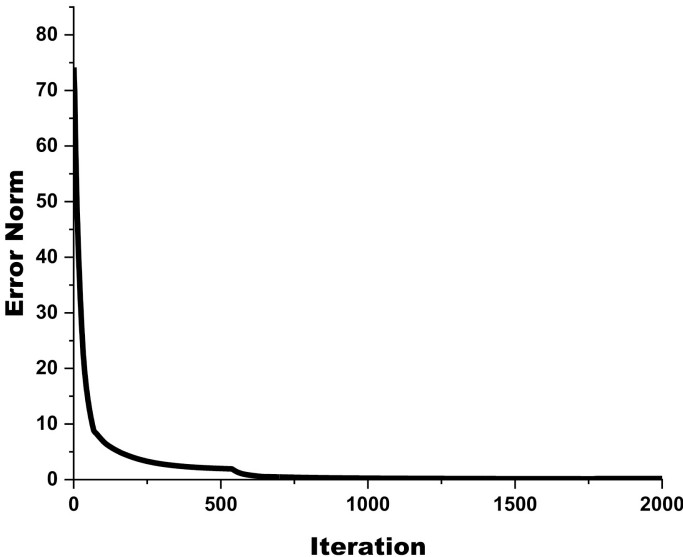

**Figure 7.** The error norm of the ANN learning of sensor (b) in Table 1.

**Table 2.** The confusion matrix for the machine utilization abstracting neural network.

| | | Prediction by the Neural Network Model | |
| --- | --- | --- | --- |
| | | Machine is working | Machine is not working |
| **Reality** | Machine is working | True positive (TP) | False negative (FN) |
| | Machine is not working | False positive (FN) | True negative (TN) |

Four time periods were selected from the IoT database, as listed in Table 3. All the time slots lasted one hour. The time periods 1 and 2 are continuous hours on the same working day, and time periods 3 and 4 are the first working hour on different working days. The ideal data count for each sensor is 3600 (1 Hz resolution), the signal counts of all time periods listed in Table 3 are close to the ideal data count. The prediction accuracies of the neural network model are 100% for all time periods shown in Table 3.

**Table 3.** The validation results for the machine utilization abstracting neural network.

| Time Period | Sensor/Machine Number [1] | Amount of Signals |
| --- | --- | --- |
| 1 | (a) | 3311 |
| | (b) | 3514 |
| | (c) | 3505 |
| | (d) | 3514 |
| 2 | (a) | 3491 |
| | (b) | 3465 |
| | (c) | 3523 |
| | (d) | 3496 |

**Table 3.** *Cont.*

| Time Period | Sensor/Machine Number [1] | Amount of Signals |
|:---:|:---:|:---:|
| 3 | (a) | 3570 |
| | (b) | 3479 |
| | (c) | 3466 |
| | (d) | 3451 |
| 4 | (a) | 3352 |
| | (b) | 3425 |
| | (c) | 3343 |
| | (d) | 3401 |

[1] The numbering of the sensor/machine follows Table 1.

### 4.3. The Collaboration Analysis between Workstations

Since the ANN model can abstract the utilization of the workstation, which is manually validated by 55 k data pairs (Table 3), the collaboration between the workstations can then be studied. The validated sensors (a)–(d) were physically located on the same production line, and the utilization has been plotted against the time.

The ideal collaboration between machines (a)–(d) was designed based on the manufacturing sequence and characteristics considering production line balance based on the throughput time listed in the third row of Table 1. The output of the machine (a) was supplied equally to machines (b) and (d). The output of machines (b) and (d) was exchanged and then put to machine (c), as illustrated by Figure 8a.

To understand the collaboration in field application, the utilization neural network models were first applied to the sensors (a)–(d). The accumulated working/non-working count against sensors (a)–(d) was plotted from a fixed starting time, as indicated in Figure 9. Observing the occurrence of the significant non-working (negative peaks with the emphasis by grey arrows), one could identify that the collaborating sequence of machines (a)–(d) followed the (i) to (vi) sequence shown in Figure 9. The collaboration between the machines is illustrated in Figure 8b. Therefore, the IoT system reveals the actual collaboration among the workstations and provides clear evidence for production efficiency improvement.

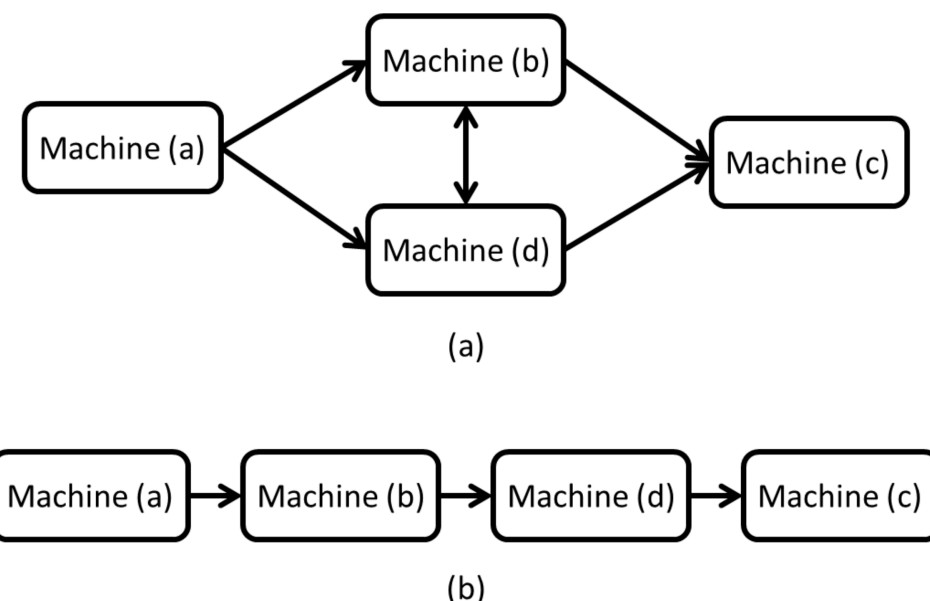

**Figure 8.** Ideal (**a**) and actual (**b**) collaboration between machines (**a**–**d**).

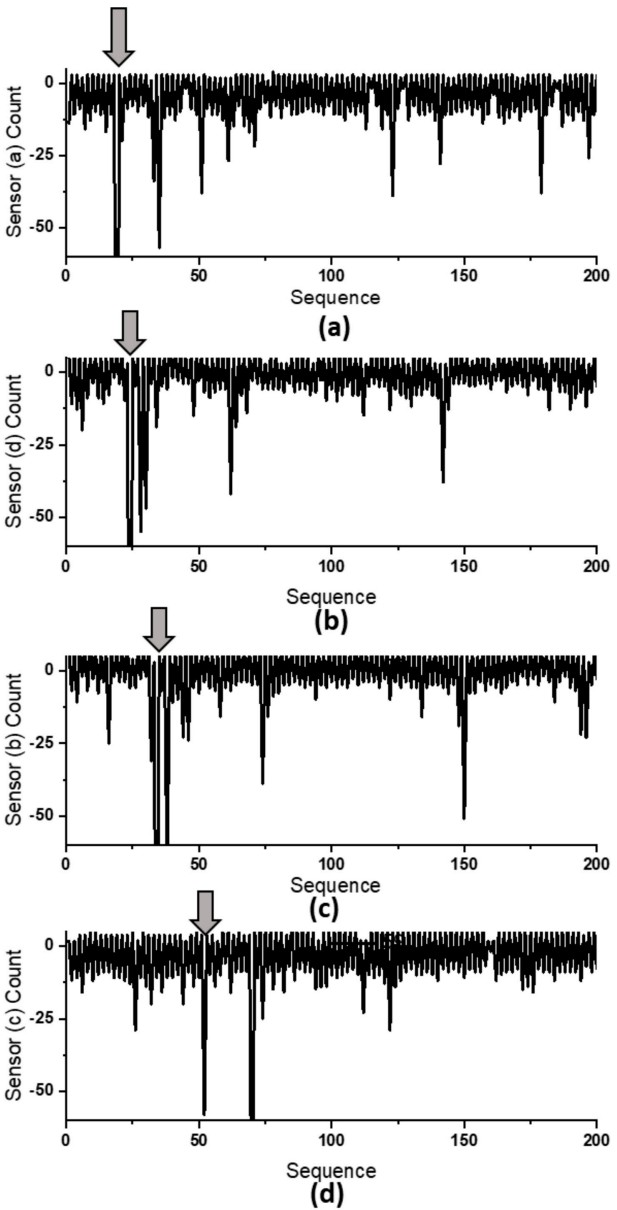

**Figure 9.** Accumulated working/non-working count for sensors (**a**–**d**).

## 5. Deep Machine Learning and Model Validation

### 5.1. Traning of the LSTM Model

As indicated in Figure 8, sensors (a)–(d) represent the activities of machines located next to each other with a fixed working sequence. The machine (b)–(d)–(c) can be considered as a production line with the input from the production result of the machine (a). Because all the machines are operated by workers, each workstation's input/output frequency is not a fixed value. Along the same sequence, Figure 10 shows the inputs by the machine (a) and the output of the machine (c). When the value is 1, it means that there is an input/output count at that second. Otherwise, there is no input/output during that particular timespan. All the data are based on the field IoT raw data and abstracted by the machine utilization neural network shown previously. From Figure 10, it is clear to see that the frequencies of input/output vary by time, and it is difficult for industries to predict the line production per day, even though the IoT system has digitally recorded the actual manufacturing process.

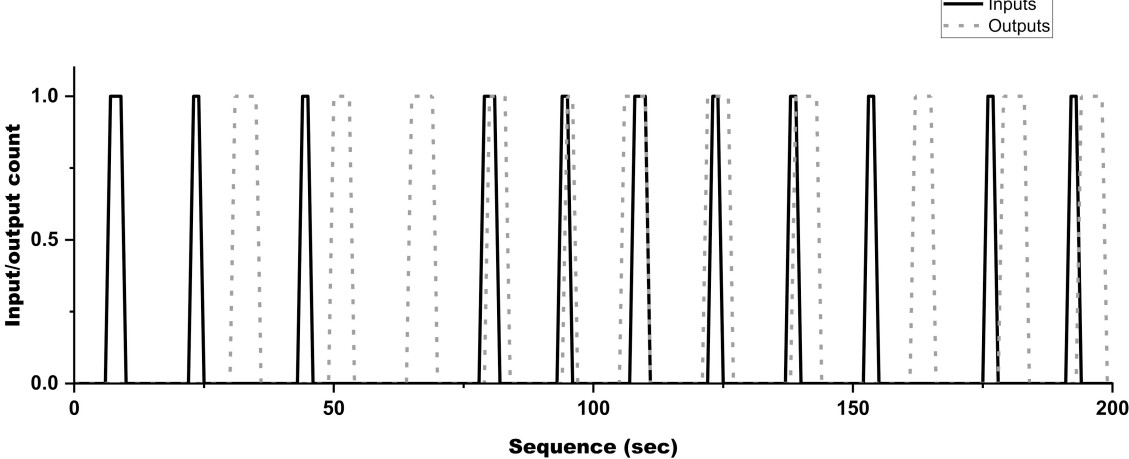

**Figure 10.** The inputs by machine (a) and outputs from the machine (c).

Hence, we applied the AI-based sequential neural network to build the virtual manufacturing model in this research. During the machine training, this model abstracted the human activities and their interactions by IoT data. Furthermore, the gnLSTM architecture, as indicated in Figure 4, was applied because gnLSTM provides sufficient flexibility to capture the field manufacturing reality by the neural work gates.

The design of the gate-networks of gnLSTM are listed in Table 4. The LSTM architecture of Figure 4 can be fulfilled by applying Equation (2). The structures of these gate-networks are listed in Table 4. The five inputs are the input from the machine (a), initial stocks, output, machine status, and stock during operation. The input from the machine (a) is obtained from the IoT system. The output, machine status, and stock during operation are the three outputs from each sequential iteration, and these three parameters are updated and become part of the inputs for the following computation. In summary, the gnLSTM is used to represent the production sequence of machines (b)–(d)–(c) that is illustrated in Figure 8b. By testing the gate network structure, activation function, learning speed with the balance of the training speed, accuracy, and avoidance of the overfitting error, the optimized configuration is reported in Table 4. Furthermore, the overfitting can be controlled because the gnLSTM structure is small compared to the size of the training dataset.

**Table 4.** The gate-network for gnLSTM and learning parameters.

| Gates | Activation Gate | Input Gate | Forget Gate | Output Gate |
|---|---|---|---|---|
| Neural network structure | (5,7,3) | (5,7,7,3) | (5,7,3) | (5,7,7,7,7,3) |
| Activation Function | Sigmoid | ReLU | Tanh | ReLU |
| Learning Speed | 0.1 | 0.015 | 0.1 | 0.015 |
| Optimizer | BPTT [1] | Adam [2] | BPTT [1] | BPTT [1] |

[1] BPTT stands for backpropagation through time optimizer. [2] Adam stands for adaptive moment estimation optimizer.

To train the gnLSTM model properly, the quality of the training set is essential. There are two key parameters, including the sequences for each training set and the amount of training sets. Because each training set should cover at least one input/output and not take too long to keep reasonable training time, each training set is fixed to 30 data pairs

by considering signal characteristics of Figure 10. The amount of the datasets was set to 30 because it is equivalent to the 15 min data if the IoT data frequency is 1 Hz.

Different activation functions were applied to gnLSTM, as listed in Table 4. The selection of the activation functions depended on many trial-and-error approaches and one combination was found to exhibit better convergence. Owing to the characteristics of the gnLSTM learning [18], these four gate-networks were optimized independently, including the learning rate and optimizers. The learning parameters of the gate-networks are listed in Table 4. Random weighting was applied in the research. However, to stabilize the training of gnLSTM, an optimized initial weighting was obtained by the genetic algorithm (GA) [20]. Figure 11 shows the convergence curve of the gnLSTM training, where a rapid reduction of the error L2Norm is depicted. It is also shown in previous gnLSTM studies [18,19].

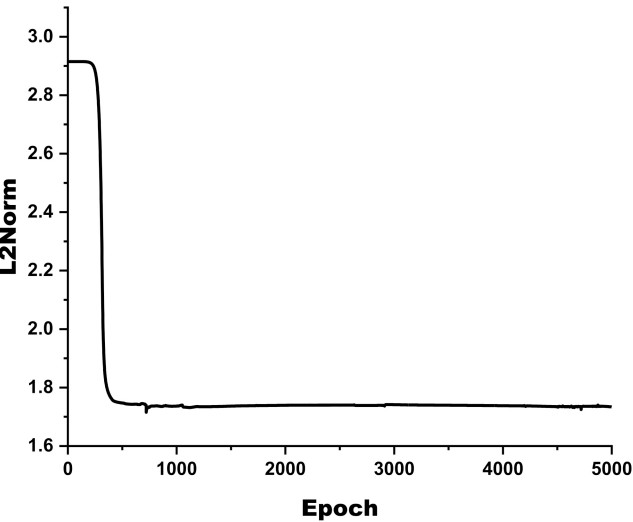

**Figure 11.** The converge norm of the gnLSTM model.

### 5.2. Validating the AI Model

To validate the gnLSTM model, the IoT information was selected to fulfill less than 5% of the data-missing rule. First, the gnLSTM model was trained using the first 15 min IoT data on day X. Afterward, the next half hour, one hour, and two hours of data were retrieved from the field and compared with the model prediction, as listed in Table 5. Although the error propagates with the expansion of the duration, the error percentage was still within the 10% error window. If the whole working day iwass considered, the prediction accuracy of day X reached −0.84%. The prediction capability against the previous working day data was 2.67% error. Days X + 1 and X + 2 were non-working days (the weekend), but the X + 3 day showed a significant error (−12.47%).

**Table 5.** The prediction accuracy of the gnLSTM model with a 15 min data training on day X.

| Day | Duration (hs) | Output, Field Reported | Output, Model Predicted | Error | Error Percentage (%) |
|---|---|---|---|---|---|
| X | 0.5 | 107 | 107 | 0 | 0.00% |
| X | 1 | 202 | 204 | −2 | −0.99% |
| X | 2 | 364 | 390 | −26 | −7.14% |
| X | 12 | 1658 | 1672 | −14 | −0.84% |
| X − 1 | 12 | 1486 | 1445 | 41 | 2.76% |
| X + 3 | 12 | 1820 | 2047 | −227 | −12.47% |

Next, the gnLSTM model was re-trained using the first 15 min of IoT data from Day X + 3 (which is equal to Day Y in Table 6), and the IoT data of the following four working days were collected and compared with the gnLSTM model, they showed an error within 5%, as listed in Table 6.

To verify our findings, the data of 8 days after day Y (which is equal to Day Z in Table 7) was applied to train the gnLSTM model. An averaged absolute error of 5% was obtained, as indicated in Table 7.

Notably, the AI training of the gnLSTM required approximately 12 min on a commercially available 12 core CPU desktop without GPU acceleration, which is affordable by most manufacturing companies. Using the combination of the IoT architecture and AI model, the manufacturing managers can reasonably predict the 12 h (current day) output by using the first 10 min of information. Moreover, production planners can use the weekly production prediction to fulfill the customers' purchasing orders.

**Table 6.** The prediction accuracy of the gnLSTM model with a15 min data training on day Y.

| Day | Output, Field Reported | Output, Model Predicted | Error | Error Percentage (%) |
|-----|------------------------|-------------------------|-------|----------------------|
| Y | 1820 | 1902 | −82 | −4.51% |
| Y + 1 | 1905 | 1915 | −10 | −0.52% |
| Y + 2 | 1679 | 1689 | −10 | −0.60% |
| Y + 3 | 1658 | 1667 | −9 | −0.54% |

**Table 7.** The prediction accuracy of the gnLSTM model with 15 min data training on day Z.

| Day | Output, Field Reported | Output, Model Predicted | Error | Error Percentage (%) |
|-----|------------------------|-------------------------|-------|----------------------|
| Z | 1503 | 1511 | −8 | −0.53% |
| Z + 1 | 1713 | 1813 | −100 | −5.84% |
| Z + 2 | 1612 | 1778 | −166 | −10.30% |
| Z + 3 | 1300 | 1296 | 4 | 0.31% |

## 6. Conclusions

Field manufacturing information acquisition and production prediction by the IoT architecture and AI modeling raises attention due to global market uncertainty and rapid production changes.

In this research, an IoT system was designed and implemented in the mass-production site of the labor-intensive shoe-making factory to monitor the workers' activities. The vibrating sensor module was installed on the workstations with limited influence on the workers' daily work. The information was filtered, abstracted, time-labeled, and then transmitted by the sensor module. The installation of the sensor was fine-tuned to minimize the transmission frequency with an acceptable signal/noise ratio. Within a specific period, the packaged information was wirelessly transmitted to the cloud. During the first month after IoT implementation, around half of the working days suffered from more than 5% missing data, mainly due to the factory-to-cloud communication issue.

Based on neural network architecture, an AI machine utilization prediction model was applied in this research to predict the machine status due to the missing data issue. The training results show that this AI machine utilization model is excellent for the single workstation status prediction for a given sewing pattern.

Considering a production line composed of multiple workstations, the actual collaboration among them can be revealed by the clean IoT data, and it can be valuable for the improvement of industrial engineering, such as the lean production concept. Moreover, the uncertainty of the labor-intensive workstation shows less fixed output frequency, which

fails to provide a reliable production prediction. A gnLSTM was carefully designed to learn the characteristics of the workstations. This research shows that the gnLSTM model trained by the first 15 min of data can predict the current and next few days' productions within 10% error. Combining the IoT architecture and AI model, it can be expected that the manufacturing site can reasonably predict the 12 h (current day) output by using the first 15 min of information. Moreover, industrial production planners can use the weekly production prediction to fulfill the customers' purchasing orders.

**Author Contributions:** Conceptualization, methodology, AI software, formal analysis, investigation, data curation, writing—original draft preparation, C.Y.; IoT architecture design, firmware, installation, C.-C.W.; IoT architecture implementation at the manufacturing site and signal analysis, M.-L.C., W.-T.L. and P.-A.L.; Data cleaning, machine utilization abstracting, gnLSTM model training, and data analysis, C.-C.L. and Z.-L.T. All authors have read and agreed to the published version of the manuscript.

**Funding:** This research is partially supported by the "Research of the industrial internet of things for the continuous manufacturing with heterogeneous machines, the production status prediction model and the smart decision-making algorithm" project of Feng Chia University, sponsored by the Ministry of Science and Technology, under the grand no. MOST 109-2221-E-035-008-MY2.

**Data Availability Statement:** The data presented in this research study are available in this article.

**Conflicts of Interest:** The authors declare no conflict of interest.

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
