# Peer review of "Using a Flexible IoT Architecture and Sequential AI Model to Recognize and Predict the Production Activities in the Labor-Intensive Manufacturing Site"

_electronics, doi:10.3390/electronics10202540_

Round 1

Reviewer 1 Report

  1. The author proposed a novel AI approach for IoT application in real-time.
  2. In Introduction, its good to mention the impact of different technology connection for wireless transmission for example: Wi-Fi, Private Cellular (CBRS), etc
  3. The impact of wireless connection to predict the performance will be also more interesting

Reviewer 2 Report

This paper proposes an IoT architecture for data collection and cloud-based data storage for a labor-intensive industry, such as a shoe-making factory. Using IoT data, two AI models for labor utilization and production prediction are designed to monitor the workers' activities. 

Suggestions for improvements:

- you mentioned that "The rectified linear unit (ReLU) is applied as the activation function."

why? elaborate more on why this function was chosen

- same goes for softmax function. You mentioned that "A softmax function is applied to determine the machine's working status."

why? Give more details on why softmax is important for predicting a probability distribution.

- besides stating that the gnLSTM "frees the LSTM gates from scalar to vector", give more details on how it behaves.

e.g. how does gnLSTM deal with the problem of vanishing gradients?

do you need to apply random weight initialization and how does it affect your LSTM?

did you experience some overfitting, give more details on the dropout/regularization procedure.

- what represents an "ideal data count"?

minor fixes:

- use consistent writing (IOT or IoT)
- fix keyword: "Could database"
- missing space in "Jazdi[8]"
- fix "cloud base database" -> "cloud-based database" in Fig. 1 caption
